Protein-RNA linkage and posttranslational modifications of feline calicivirus and murine norovirus VPg proteins

Olspert Allan 1
Hosmillo Myra 2
Chaudhry Yasmin 2
Peil Lauri 3
Truve Erkki 1
http://orcid.org/0000-0002-9483-510X Goodfellow Ian 2 ig299@cam.ac.uk
1 Faculty of Science, Department of Gene Technology, Tallinn University of Technology , Tallinn , Estonia
2 Division of Virology, Department of Pathology, University of Cambridge , Cambridge , United Kingdom
3 Faculty of Science and Technology, Institute of Technology, University of Tartu , Tartu , Estonia
Grande-Pérez Ana
Electronic publication date: 2016 Jun 28
Publication date: 2016
Volume: 4
Electronic Location ID: e2134
Received 2016 Apr 12; Accepted 2016 May 24
Copyright: © 2016 Olspert et al.
Copyright year: 2016
Copyright holder: Olspert et al.
License: This is an open access article distributed under the terms of the Creative Commons Attribution License, which permits unrestricted use, distribution, reproduction and adaptation in any medium and for any purpose provided that it is properly attributed. For attribution, the original author(s), title, publication source (PeerJ) and either DOI or URL of the article must be cited.
License URL: https://creativecommons.org/licenses/by/4.0/

Keywords: Protein-RNA linkage, Posttranslational modifications, Calicivirus, Feline calicivirus, Murine norovirus, VPg

Funding: Funding is provided by Wellcome Trust (Ref: 097997/Z/11/Z) and by the institutional research funding IUT 19-3 of the Estonian Ministry of Education and Research. The funders had no role in study design, data collection and analysis, decision to publish, or preparation of the manuscript.

==============================
Members of the Caliciviridae family of positive sense RNA viruses cause a wide range of diseases in both humans and animals. The detailed characterization of the calicivirus life cycle had been hampered due to the lack of robust cell culture systems and experimental tools for many of the members of the family. However, a number of caliciviruses replicate efficiently in cell culture and have robust reverse genetics systems available, most notably feline calicivirus (FCV) and murine norovirus (MNV). These are therefore widely used as representative members with which to examine the mechanistic details of calicivirus genome translation and replication. The replication of the calicivirus RNA genome occurs via a double-stranded RNA intermediate that is then used as a template for the production of new positive sense viral RNA, which is covalently linked to the virus-encoded protein VPg. The covalent linkage to VPg occurs during genome replication via the nucleotidylylation activity of the viral RNA-dependent RNA polymerase. Using FCV and MNV, we used mass spectrometry-based approach to identify the specific amino acid linked to the 5′ end of the viral nucleic acid. We observed that both VPg proteins are covalently linked to guanosine diphosphate (GDP) moieties via tyrosine positions 24 and 26 for FCV and MNV respectively. These data fit with previous observations indicating that mutations introduced into these specific amino acids are deleterious for viral replication and fail to produce infectious virus. In addition, we also detected serine phosphorylation sites within the FCV VPg protein with positions 80 and 107 found consistently phosphorylated on VPg-linked viral RNA isolated from infected cells. This work provides the first direct experimental characterization of the linkage of infectious calicivirus viral RNA to the VPg protein and highlights that post-translational modifications of VPg may also occur during the viral life cycle.

Introduction

The RNA genomes of several positive sense RNA viruses are covalently linked to virus-encoded protein, referred to as VPg. Vertebrate RNA viruses that encode a VPg protein include picornaviruses, astroviruses and caliciviruses (Goodfellow, 2011). In picornaviruses, VPg is a 22 amino acid peptide that is linked to the 5′ end of the RNA via a phosphodiester bond between the hydroxyl group of a tyrosine residue and the 5′ phosphate group of the viral genomic RNA, which invariably starts with a pUpU sequence (Ambros & Baltimore, 1978; Rothberg et al., 1978). The linkage of VPg to picornavirus RNA occurs during viral genome replication in a process whereby the viral RNA-dependent RNA polymerase uses an RNA structure as a template for the addition of a pUpU moiety to a highly conserved tyrosine residue within the VPg peptide (Hewlett & Florkiewicz, 1980; Goodfellow et al., 2000; Paul et al., 2000). The VPg protein of caliciviruses and astroviruses is typically 13–15 kDa in size and is essential for the infectivity of viral RNA (Goodfellow et al., 2005; Chaudhry et al., 2006; Fuentes et al., 2012; Hosmillo et al., 2014). In the case of feline calicivirus (FCV) and murine norovirus (MNV), VPg plays an essential role in viral protein synthesis that has been linked to the ability of VPg to interact with cellular translation initiation factors, eIF4E in the case of FCV and eIF4G in the case of MNV (Goodfellow et al., 2005; Chaudhry et al., 2006; Chung et al., 2014). In plants, members of Secoviridae, Potyviridae, Luteoviridae families and Sobemovirus genus possess VPg proteins of 2–24 kDa in size, covalently linked to the 5′-terminal nucleotide of the viral RNA via tyrosine, serine or threonine residues within VPg (Jiang & Laliberté, 2011; Olspert et al., 2011a; Olspert et al., 2011b). Whilst VPg of plant viruses play multiple roles in the viral life cycle, the best characterized potyvirus VPg interacts with the canonical factor eIF4F also confirming a role in virus translation and the regulation of host gene expression (Jiang & Laliberté, 2011).

The covalent linkage of VPg to the 5′ end of calicivirus RNAs has previously been examined by iodination of purified viral RNA, confirming that VPg is linked to both genomic and subgenomic viral RNAs (Herbert, Brierley & Brown, 1997). Reverse genetics has also been used to identify the key amino acids in VPg required for viral infectivity of both FCV and MNV (Mitra, Sosnovtsev & Green, 2004; Subba-Reddy, Goodfellow & Kao, 2011; Leen et al., 2013); however, the multifunctional role of VPg has complicated the direct identification of amino acids involved in linkage to viral RNA as well as the precise nature of the nucleotide that is linked. Therefore, direct experimental confirmation of the amino acid-RNA linkage to infectious calicivirus RNA is still lacking.

Here we have used mass-spectrometry based characterization to identify the amino acids involved in VPg-linkage to viral RNA in both FCV and MNV, demonstrating a direct linkage to pGp moieties. In addition, we identified the possible posttranslational modifications that may contribute to the regulation of VPg function during the calicivirus life cycle.

Materials and Methods

Virus culture and isolation of viral VPg-linked RNA

The F9 strain of FCV and the CW1 isolate of MNV were grown in Crandell-Reese feline kidney cells (obtained from ATCC) and RAW264.7 cells (obtained from ATCC) respectively. For each preparation, at least five 170 cm2 flasks were infected with a multiplicity of infection of 0.2 TCID50/cell. Infected cells were harvested at ∼15 h post infection. Cells were resuspended directly in lysis buffer and the total RNA was isolated using the GenElute mammalian total RNA miniprep kit as per the manufacturer’s instructions. Eluted samples were combined, further concentrated by ethanol precipitation and resuspended in nuclease free water. Where required, preparations of VPg-linked RNA were treated with RNase cocktail (Ambion) at 37 °C for 1 h prior to the addition of SDS-PAGE sample buffer and separation by 15% SDS-PAGE.

Recombinant protein expression and purification

Untagged derivatives of FCV and MNV VPg proteins were expressed and purified in E. coli as previously described (Goodfellow et al., 2005; Chaudhry et al., 2006).

Mass-spectrometric analysis of FCV and MNV VPg-linked RNA

VPg, covalently bound to the RNA, was trypsin digested and the RNA was subsequently hydrolyzed in 10% trifluoroacetic acid for 48 h at room temperature. 10 μg of total RNA, measured by an absorbance-based quantitation (NanoDrop), was used per analysis. The samples were then dried under vacuum, purified with StageTips (Rappsilber, Mann & Ishihama, 2007) and analyzed by LC–MS/MS using an Agilent 1,200 series nanoflow system (Agilent Technologies) connected to a LTQ Orbitrap mass-spectrometer (Thermo Electron) equipped with a nanoelectrospray ion source (Proxeon), as described previously (Olspert et al., 2011a). LTQ Orbitrap was operated in the data dependent mode with a full scan in the Orbitrap (mass range m/z 300–1,900, resolution 60,000 at m/z 400, target value 1 Â 106 ions) followed by up to five MS/MS scans in the LTQ part of the instrument (normalized collision energy 35%, wideband activation enabled, target value 5,000 ions). Fragment MS/MS spectra from raw files were extracted as MSM files and then merged to peak lists using Raw2MSM version 1.11, selecting top eight peaks for each 100 Da (Olsen et al., 2005). MSM files were searched with the Mascot 2.3 search engine (Matrix Science) against the protein sequence database composed of VPg sequences and common contaminant proteins such as trypsin, keratins etc. Search parameters were as follows: 5 ppm precursor mass tolerance and 0.6 Da MS/MS mass tolerance, three missed trypsin cleavages plus a number of variable modifications such as oxidation (M), oxidation (HW), ethyl (DE), phospho (ST), phospho (Y), pAp (STY), pGp (STY), pCp (STY) and pUp (STY). For both viruses at least two independent biological samples were analyzed. For publication the spectra were auto-annotated with xiSPEC (http://spectrumviewer.org) and images were prepared using Inkscape (http://www.inkscape.org). The levels of viral RNA present in the samples used for mass spec were not routinely quantified, however yields from identical preparations were typically in the range of ∼107 genome equivalents per μg of total RNA.

Results and Discussion

Conservation of the calicivirus 5′ end and VPg sequences

We initially compared the VPg sequences from a number of representative caliciviruses to identify amino acids that were highly conserved across the genera (Fig. 1A). Calicivirus VPg sequences vary in length from 65 amino acids for bovine nebovirus to 138 amino acids for Norwalk virus. The recent solution of the structures of the VPg proteins from MNV, FCV, and porcine sapovirus (PSaV) highlight the presence of conserved helical bundles at the core of VPg, tightly bound in hydrophobic interactions (Leen et al., 2013; Hwang et al., 2015). A limited number of amino acids were highly conserved across all calicivirus genera, most notably a lysine rich N-terminal region and a central motif containing the EYDEΦ sequence, with Φ representing any aromatic amino acid (Fig. 1A). The tyrosine within this motif, position 24 and 26 of FCV and MNV respectively, has previously been proposed as a possible site for VPg nucleotidylylation based on data using either in vitro biochemical assays (Machín, Martín Alonso & Parra, 2001; Belliot et al., 2008; Han et al., 2010) or a novel cell-based assay (Subba-Reddy, Goodfellow & Kao, 2011). However, discordant data was obtained for MNV where in vitro biochemical assays identified tyrosine at position 117 as the site for nucleotidylylation (Han et al., 2010). Importantly, mutational analysis of this amino acid in the context of the MNV infectious clone confirmed that Y117 was not required for viral infectivity whereas Y24 was essential (Subba-Reddy, Goodfellow & Kao, 2011). This highlights that, at least for MNV, in vitro biochemical approaches using recombinant purified proteins, are confounded by an apparent lack of specificity of the viral RdRp.

Figure 1 Comparison of calicivirus VPg and 5′ genomic and subgenomic sequences.

(A) Amino acid alignment of VPg sequences among representatives of calicivirus genera: MNV (DQ285629), Norwalk (AF093797), FCV (M86379), PSaV (AF182760), RHDV (Z49271), Tulane (EU391643) and BEC-NB (AY082891). The conserved amino acids are coloured including the highly conserved central motif of VPg, EYDEΦ (Φ is any aromatic acid). An asterisk (*) indicates the conserved tyrosine (Y) residue essential for calicivirus replication. A hash (#) indicates the Y residue identified necessary for MNV nucleotidylylation using an in vitro[i] biochemical approach (Han et al., 2010). The identified phosphorylation (Phos) sites in the FCV VPg protein are shaded. Alignment of the first 20 nucleotides of the genomic (B) and (C) subgenomic RNAs of representative caliciviruses. The putative VPg-linked 5′ G nucleotides are highlighted and shown in red. AUG are shown in blue. The solution structure of the FCV (D, PDB: 2M4H) and MNV (E, PDB: 2MG4) VPg proteins are also shown. The FCV structure represents amino acid G10 to Y76 whereas the MNV VPg structure encompasses amino acids G11 to L85.

Alignment of the 5′ end sequences of representative calicivirus genomes has demonstrated that almost all genomic and subgenomic RNAs start with a GU dinucleotide (Figs. 1B and 1C). Tulane virus is an exception to this, as published data would indicate that the genome begins with a GGG sequence (Farkas et al., 2008). It is worth noting that there is only a single full-length genome sequence available for Tulane virus, therefore confirmation of the 5′ end may require additional viral sequences. Based entirely on sequence conservation, we would expect calicivirus VPg proteins to be guanylylated on the conserved tyrosine within the EYDEΦ sequence equivalent to positions 24 and 26 for FCV and MNV respectively. These amino acids are predicted to lie within the structured region of the FCV and MNV proteins as highlighted in Figs. 1D and 1E, respectively.

Purification of viral VPg-linked RNA

In order to identify the nucleotide and the amino acid involved in the covalent linkage of calicivirus RNA to VPg, a source of viral VPg-linked RNA that would yield sufficient nucleotide-linked VPg was required. Attempts were initially made to purify sufficient quantities of VPg-linked viral RNA from infectious virions, however the yields were insufficient to allow the robust detection of VPg peptides by mass spectrometry. As an alternative approach, we isolated total RNA from infected cells as we have previously demonstrated that RNA isolated in this way is covalently linked to the mature form of VPg only, that the RNA is infectious when transfected into permissive cells, that the VPg-linked RNA is translationally competent and that the linkage to VPg is essential for the infectivity of the viral RNA (Goodfellow et al., 2005; Chaudhry et al., 2006; Hosmillo et al., 2014). Taken together, these observations confirm that RNA isolated from infected cells provides a robust source of authentic VPg-linked viral RNA. Cells permissive for either FCV or MNV infection were infected with a low multiplicity of infection and total RNA purified using a silica column based purification method. To determine if sufficient quantities of VPg-linked RNA were present within these preparations, the RNA was digested with a cocktail of ribonuclease A and T1, separated by SDS-PAGE and proteins visualised by staining with colloidal Coomassie blue (Fig. 2). Recombinant VPg proteins expressed and purified from E. coli without any exogenous non-viral amino acids were used to confirm the expected mass of the VPg proteins. FCV VPg was readily visible following Coomassie staining of RNase digested RNA; however MNV VPg, because of its larger mass, was obscured by the RNase present within the sample. Western blot analysis, and subsequent mass spectrometry (see below) was used to confirm the isolation of the MNV VPg protein.

Figure 2 Isolation and characterization of calicivirus VPg-linked RNA.

Total RNA was isolated from FCV, MNV or mock-infected cells then ∼10 μg was subjected to RNase treatment. The calicivirus VPg linked to the RNA were subsequently analysed in SDS-PAGE, alongside their corresponding recombinant proteins. White arrowheads indicate the recombinant VPg used as a marker with black arrowhead indicating the position of VPg linked to the RNA. An asterisk (*) is used to highlight the position of the RNase A (13.7 kDa) in the treated samples.

Detection of FCV and MNV VPg peptides and post-translational modifications

Initial attempts were made to analyse the RNase treated and trypsin-digested VPg-linked RNA preparations as a method to identify the nucleotide and amino acids involved in the covalent linkage; however, this approach failed to produce spectra that allowed for the detection of the nucleotide-linked VPg peptides. As an alternative approach, we used tryptic digestion of VPg-linked RNA preparations followed by acid hydrolysis of the RNA-linked peptides as described previously (Olspert et al., 2011a; Olspert et al., 2011b). The RNA-linked amino-acid residue modification after RNA hydrolysis using this method is known to be a 5′,3′-diphosphate nucleotide, pNp (N denoting adenosine, cytidine, guanosine or uridine), and the possible phosphodiester bond acceptor residues are serine, tyrosine and threonine. FCV and MNV VPg-linked RNA preparations were prepared and analyzed by Orbitrap MS. The sequence coverage obtained for the FCV and MNV VPg proteins were 70 and 63%, respectively. The identified peptides are shown in Table 1, Figs. 3 and 4. The regions not detected were most likely absent due to trypsin digestion producing peptides too short for detection or confident identification.

Table 1 Examples of detected peptides identified by fragmentation spectra.

The post-translational modifications are described and the modified position is in bold in the peptide.

Virus	Position	Peptide	Modification	Experimental mass, Da	Calculated mass, Da	Mass error, ppm	Mascot score	
FCV	9–13	IGTYR	phos	688.2952	688.2945	1.05	20	
FCV	9–13	IGTYR		608.329	608.3282	1.39	32	
FCV	16–28	GVALTDDEYDEWR	GDP	1,992.6959	1,992.6928	1.54	61	
FCV	16–28	GVALTDDEYDEWR	GDP, ox	2,008.6909	2,008.6877	1.56	52	
FCV	16–28	GVALTDDEYDEWR	GDP, eth	2,020.7274	2,020.7241	1.62	50	
FCV	16–28	GVALTDDEYDEWREHNASR	GDP	2,687.0071	2,687.0075	−0.16	10	
FCV	35–47	KLDLSVEDFLMLR	ox	1,593.8448	1,593.8436	0.77	86	
FCV	36–47	LDLSVEDFLMLR	ox	1,465.7502	1,465.7487	1.09	68	
FCV	50–61	AALGADDNDAVK	eth	1,186.5837	1,186.583	0.62	76	
FCV	50–61	AALGADDNDAVK		1,158.5542	1,158.5517	2.23	63	
FCV	64–69	SWWNSR	ox	850.3722	850.3722	0.066	29	
FCV	64–69	SWWNSR	ox, ox	866.3674	866.3671	0.39	16	
FCV	64–69	SWWNSR		834.3776	834.3773	0.45	15	
FCV	72–84	MANDYEDVTVIGK	ox	1,469.672	1,469.6708	0.84	96	
FCV	72–84	MANDYEDVTVIGK	ox, phos	1,549.6372	1,549.6371	0.076	65	
FCV	72–84	MANDYEDVTVIGK		1,453.6756	1,453.6759	−0.17	111	
FCV	103–111	GYDVSFAEE	phos	1,095.3796	1,095.3798	−0.12	31	
FCV	103–111	GYDVSFAEE		1,015.4134	1,015.4135	−0.0039	42	
MNV	9–17	GRPGVFR		787.4444	787.4453	−1.1	32	
MNV	20–30	GLTDEEYDEFK	GDP	1,769.5878	1,769.5859	1.05	36	
MNV	20–30	GLTDEEYDEFK		1,344.572	1,344.5721	−0.13	64	
MNV	20–31	GLTDEEYDEFKK	GDP	1,897.6834	1,897.6808	1.36	29	
MNV	20–31	GLTDEEYDEFKK		1,472.6673	1,472.6671	0.13	25	
MNV	40–49	YSIDDYLADR	eth	1,257.5883	1,257.5877	0.48	70	
MNV	40–49	YSIDDYLADR		1,229.5588	1,229.5564	1.92	52	
MNV	40–51	YSIDDYLADRER		1,514.6998	1,514.7001	−0.19	23	
MNV	50–58	EEELLER		916.4508	916.4501	0.72	19	
MNV	50–58	EREEELLER		1,201.5946	1,201.5938	0.64	63	
MNV	50–72	EREEELLERDEEEAIFGDGFGLK	eth	2,737.3105	2,737.3082	0.82	11	
MNV	50–72	EREEELLERDEEEAIFGDGFGLK		2,709.2807	2,709.2769	1.39	52	
MNV	52–72	EEELLERDEEEAIFGDGFGLK		2,424.1361	2,424.1332	1.2	77	
MNV	59–72	DEEEAIFGDGFGLK		1,525.6944	1,525.6936	0.48	103	
MNV	85–94	LGLVSGGDIR	eth	1,013.5865	1,013.5869	−0.42	42	
MNV	85–94	LGLVSGGDIR		985.5566	985.5556	0.92	55	
MNV	97–113	KPIDWNVVGPSWADDDR		1,968.9345	1,968.933	0.75	33	
Note:

phos, phosphorylation; ox, oxidation; eth, ethylation; GDP, guanosine diphosphate; pGp.

Figure 3 Mass-spectrometric characterization of FCV VPg.

(A) FCV VPg, regions for which peptides were detected are shown in red, the amino acid residue linked to RNA is in blue and underlined, phosphorylated residues are in bold green, the regions not detected are in lowercase black. (B–D) Identification of post-translational modifications of FCV VPg by MS/MS analysis. Co-purified VPg linked to RNA was trypsin-digested and RNA was degraded with acidic hydrolysis. The peptides were analyzed by nano-LC/MS/MS and resulting data was searched against corresponding sequence databases by MASCOT. The a/b and y ions represent peptide N- and C-terminal fragment ions produced by collision-induced dissociation in the mass spectrometer. Matched ions are indicated in red and corresponding losses indicated at the top left corner of each plot, M denotes the precursor peptide (with corresponding losses). (B) Identification of the residue covalently linked to RNA. The FCV VPg peptide, GVALTDDEYDEWR, was identified to contain a Y9 linked pGp modification (indicated with GDP), a corresponding degradation product of viral RNA. (C–D) VPg peptide GYDVSFAEE was detected to contain a phosphorylation at S5 (indicated with PH) and peptide MANDYEDVTVIGK was detected to be phosphorylated at T9 (in addition to occasional M1 oxidation (OX) occurring during sample handling).

Figure 4 Mass-spectrometric characterization of MNV VPg.

(A) MNV VPg, regions for which peptides were detected are shown in red, the amino acid residue linked to RNA is in blue and underlined, the regions not detected are in lowercase black. (B) Determination of the residue covalently linked to MNV RNA by MS/MS analysis. Co-purified VPg linked to RNA was trypsin-digested and RNA was degraded with acidic hydrolysis. The peptides were analyzed by nano-LC/MS/MS and resulting data was searched against corresponding sequence databases by MASCOT. The b and y ions represent peptide N- and C-terminal fragment ions produced by collision-induced dissociation in the mass spectrometer. Matched ions are indicated in red and corresponding losses indicated at the top left corner of each plot, M denotes the precursor peptide (with corresponding losses). The MNV VPg peptide, GLTDEEYDEFK, was identified to contain Y7 linked pGp modification (indicated with GDP), a corresponding degradation product of viral RNA.

Using this approach we determined that the FCV VPg is linked to RNA through the tyrosine residue at position 24 (Y24) and the corresponding modification was pGp, as assigned by the modification delta mass and the corresponding fragmentation spectrum (Fig. 3B). This is in agreement with the high degree of conservation of a 5′ G nucleotide in all calicivirus genomes (Figs. 1B and 1C). The spectra were searched against all possible nucleotides (pGp, pUp, pCp and pAp) but no other matches were detected indicating that all the detected VPg peptides were derived from linkage to the positive strand of viral RNA. The corresponding FCV VPg peptide was never detected without pGp modification. In FCV VPg we also identified two potential phosphorylation sites; threonine at position 80 (Fig. 3C) and serine at position 107 (Fig. 3D) were consistently detected as phosphorylated but the same peptides were also detected without the modification. This might indicate loss of modification during sample handling and/or transient nature of the modification. Threonine at position 11 was also detected in a phosphorylated form (Table 1), however this was identified in only one of the two FCV samples. Unfortunately, due to peptide’s short length and low amount of matched fragmentation ions, this is a low confidence observation. Threonine at position 11 is not conserved between FCV isolates, with all other isolated possessing a proline at this position, while T80 and S107 are 100% conserved across all FCV isolates. Amino acids T80 and S107 are in disordered region of the FCV VPg protein (Fig. 1D). It is unclear as to whether the phosphorylation would have resulted in a change in the mobility of VPg on SDS-PAGE.

The significance of VPg phosphorylation in the FCV life cycle clearly warrants further study using reverse genetics and mutagenesis. VPg is a multifunctional protein; it functions in viral RNA synthesis via an interaction with NS7, in viral protein synthesis via an interaction with translation initiation factors, and possibly in viral encapsidation via an interaction with VP1 (Goodfellow, 2011). Therefore, phosphorylation may provide a mechanism with which to regulate these roles in the virus life cycle.

The MNV VPg protein was identified as also linked to RNA through a tyrosine residue at position 26 and the modification was also pGp (Fig. 4B). Surprisingly for MNV, the corresponding peptide was also detected without the RNA modification, suggesting that RNA modification was lost during sample preparation. In addition, we detected random aspartate and glutamate ethylation, methionine and tryptophan oxidations (Table 1), which are known to be generated in vitro during sample preparation (Stadtman & Levine, 2003; Xing et al., 2008; Olspert et al., 2011a) and were therefore not considered of biological relevance. We did not observe evidence of MNV VPg phosphorylation. This difference between FCV and MNV may be due to inherent biological variation or may be a simple reflection of the lower abundance of MNV VPg in the samples analyzed and the sensitivity of the assay. Further studies are required to specifically address whether MNV VPg is post-translationally modified.

Previous studies on caliciviruses VPg nucleotidylylation have examined the ability of the viral RdRp to uridylylate VPg by leading to the formation of a VPg-pUpU(OH) moiety (Rohayem et al., 2006; Belliot et al., 2008). As all calicivirus genomic and subgenomic RNAs begin with a single G, uridylylated VPg, if formed, would be expected to prime only negative sense RNA synthesis on the 3′ poly A tail present on the viral genomic RNA. The end result would be VPg-pUpU at the 5′ end of negative sense viral RNA. Whether VPg is uridylylated during calicivirus replication remains to be determined, however our data would indicate that only guanosine diphosphate (GDP) was found linked to either the FCV or the MNV VPg proteins isolated from infected cells. Given the asymmetry of calicivirus genome replication, the number of negative sense RNA molecules present within an infected cell can be > 1,000 fold lower than the corresponding positive sense RNA molecule (Vashist, Urena & Goodfellow, 2012), which even if it were linked to uridylylated VPg, may be present in such low quantities to make detection challenging. We have previously demonstrated that the norovirus RdRp possesses the ability to initiate RNA synthesis de novo and that this activity is regulated by binding to the viral capsid protein VP1 (Subba-Reddy, Goodfellow & Kao, 2011). This has allowed us to propose a model whereby negative sense RNA synthesis occurs via a primer independent de novo mechanism of RNA synthesis but positive sense RNA synthesis occurs via a VPg-primed mechanism (Reviewed in Thorne & Goodfellow (2014)). In this model, only guanylylated VPg would be produced, fitting with our current experimental observations.

In picornaviruses, the VPg protein is removed immediately upon viral RNA release into the cytoplasm by the host cell enzyme TDP2 (Virgen-Slane et al., 2012), but the removal of the VPg protein is not essential for the replication of the incoming viral RNA (Langereis et al., 2014). In contrast, ongoing picornavirus replication may require the activity of TDP2 (Maciejewski et al., 2016). Given the absolute requirement of VPg for calicivirus genome translation we would expect that the calicivirus VPg protein remains attached to the viral RNA, at least during the initial stages of the viral life cycle. Fitting with this hypothesis, preliminary in vitro analysis would indicate that purified TDP2 is unable to cleave the MNV VPg protein from viral RNA under conditions that readily removes the poliovirus VPg (B Semler and S Maciejewski, 2016, personal communication).

In conclusion, this work has provided experimental confirmation that at least for FCV and MNV, the covalent linkage of the VPg proteins to the 5′ end of the viral genome occurs specifically via a highly conserved tyrosine residue to the 5′ G nucleotide. The identification of potential phosphorylated sites in the FCV VPg protein may provide a mechanism by which the function of VPg is temporally regulated during the viral life cycle.

Additional Information and Declarations

Competing Interests

Author Contributions

Data Deposition

The authors declare that they have no competing interests.

Allan Olspert conceived and designed the experiments, performed the experiments, analyzed the data, contributed reagents/materials/analysis tools, wrote the paper, prepared figures and/or tables, reviewed drafts of the paper.

Myra Hosmillo conceived and designed the experiments, analyzed the data, contributed reagents/materials/analysis tools, wrote the paper, prepared figures and/or tables, reviewed drafts of the paper.

Yasmin Chaudhry conceived and designed the experiments, performed the experiments, analyzed the data, contributed reagents/materials/analysis tools, wrote the paper, prepared figures and/or tables, reviewed drafts of the paper.

Lauri Peil conceived and designed the experiments, performed the experiments, analyzed the data, contributed reagents/materials/analysis tools, wrote the paper, prepared figures and/or tables, reviewed drafts of the paper.

Erkki Truve conceived and designed the experiments, analyzed the data, contributed reagents/materials/analysis tools, wrote the paper, prepared figures and/or tables, reviewed drafts of the paper.

Ian Goodfellow conceived and designed the experiments, performed the experiments, analyzed the data, contributed reagents/materials/analysis tools, wrote the paper, prepared figures and/or tables, reviewed drafts of the paper.

The following information was supplied regarding data availability:

All raw data is included in the manuscript.

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
