# Peer review of "Protein-RNA linkage and posttranslational modifications of feline calicivirus and murine norovirus VPg proteins"

_PeerJ, doi:10.7717/peerj.2134_

## Round 0.1 · original submission · Minor Revisions

· Academic Editor

Minor Revisions

Your manuscript is a significant contribution to the understanding of ssRNA(+) replication that deserves publishing. Both referees are impressed by your work and have raised minor issues that will help to improve your manuscript and need to be addressed. I will be happy to receive a revised version of your article at your convenience.

Thank you

Reviewer 1 ·

Basic reporting

This nice paper by Olspert and colleagues describes the identification of Tyr residues 24 and 26 in FCV and MNV VPgs, respectively, that are linked to viral genome.
The identification of residues 24 and 26 is in agreement with precedent studies by the same group and other laboratories which also suggested that these residues are the site for VPg nucleotylylation. Nonetheless, in vitro assays carried out by Han et al 2010 suggested that nucleotylylation by the virus polymerase occurred in a different Tyr in VPg far from residues 24/26. This study by Olspert et al is solving this controversial topic relevant to the initiation of replication in caliciviruses.


In addition to their main finding, the authors also conclude that guanidine but not other nucleotides are incorporated into the VPg, suggesting that VPg is only priming positive but not negative strand synthesis (although this needs further analysis as +/- strand ratio is largely displaced to + strands). Their study also unravel different phosphorylation sites in VPg, a very interesting aspect they barely discuss, which may be relevant to future studies on the role of cellular kinases on regulating calicivirus infection. I therefore recommend this paper for its publication in PeerJ

Experimental design

The authors make use of a very simple but elegant approach which I find very stimulating: to achieve sufficient VPg-RNA protein they follow standard protocols for RNA purification that they obtain from large scale cell culture monolayer infections. After chemical hydrolysis they are able to detect VPg proteins by SDS-PAGE. VPg-PPi purified is then trypsinised and fragments detected by mass spec.

Validity of the findings

No comments

Additional comments

Minor:

1. The authors mention that they carried out WB to confirm the identity of MNV VPg. Why they don't show that in a Figure?
2. The authors barely discuss the implication of VPg phosphorylation on calicivirus replication. Has phosphorylation been described before for VPg in other caliciviruses? Do they have any idea of whether these P sites may be regulating virus replication
3. sentence 150-151 doesn't make sense. Do you mean authentic Vpg linked-RNA?
4. 164. there’s a typo. A question mark is misplaced in this sentence

Reviewer 2 ·

Basic reporting

No comments

Experimental design

No comments

Validity of the findings

No comments

Additional comments

Although experimental design is brief and concise, results are worth being published and will contribute to the progress in the ssRNA(+) virus field.
The questions and concerns raised by this reviewer, which may improve the quality of the manuscript, are summarized below:

- Line 152. The authors state that cells were infected with a high multiplicity of infection. However, in the Materials and Methods section a MOI of 0.2 TCID50/cell is indicated. I do not think that such MOI could be considered a high MOI.
- Description on the method used to measure RNA concentration after purification from infected cells should be included. The authors indicate that 10 ug of RNA were loaded in the SDS-PAGE gel, but they do not describe how they measured this concentration. In addition, data on real time RTqPCR quantification to indicate the concentration in genome copies/ug of total RNA should be included.
- The authors find two phosphorylated sites in FCV VPg, and state that unmodified VPg was also detected by MS. Would this be seen in the SDS-PAGE as a double band? A comment on this should be included somewhere in the text.
- The authors identify phosphorylated residues in FCV VPg, but they do not find any in MNV VPG. These results should be discussed in order to indicate whether this could be due to a “sensitivity” issue of the technique, or whether it can be concluded that MNV VPG does not get phosphorylated.

Minor comments:
- Line 103: “toler-ance” should be corrected
- Line 189: The authors should indicate that phosphorylation at position 11 was identified in only one FCV sample “out of how many total samples”?
- Line 208: The authors should indicate what “GDP” stands for.
- Fig 1 legend. The authors should correct “Y27” to “Y76” in the last sentence.
- Fig 2. The amount recombinant proteins loaded in the gel should be indicated. The molecular weight or RNAse T1 should also be indicated, in order to rule out its detection in the gel.

---

## Round 0.2 · accepted · Accept

· Academic Editor

Accept

Thank you for the revised version of your manuscript and for chosing PeerJ to publish your work.